# Towards Better Orthogonality Regularization with Disentangled Norm in Training Deep CNNs

## Abstract

In addressing feature redundancy and training instability in CNNs, orthogonality regularization has emerged as a promising approach. Specifically, a variant termed kernel orthogonality regularization seeks to optimize models by minimizing the residual between kernel functions of convolutional filters and the identity matrix.

Contrary to methods that measure the kernel residual as a holistic entity, our approach introduces a tailored measure that disentangles diagonal and correlation components from the kernel matrix, thereby mitigating their mutual interference during training. Models equipped with this strict kernel orthogonality measure outperform existing methods in near-orthogonality. Notably, we observe test accuracy improvements for shallow architectures. However, as model depth increases, the efficacy of our strict kernel orthogonality approach diminishes.

Given the challenges of strict kernel orthogonality in deeper models and the inherent non-compliance of specific convolutional layers with the kernel orthogonality definition, we introduce the concept of a relaxation theory, wherein strict orthogonality is a special case. By adopting this relaxed kernel orthogonality regularization, we observe enhanced model performance in deeper architectures, suggesting it as a robust alternative to the strict counterpart.

To validate our approach's efficacy in achieving near-orthogonality and enhancing model performance, we conduct rigorous experiments with our kernel orthogonality regularization toolkit on ResNet and WideResNet in CIFAR-10 and CIFAR-100 datasets. We observe state-of-the-art gains in model performance from the toolkit and obtain more robust models with expressive features. These experiments demonstrate the efficacy of our toolkit while highlighting the often overlooked challenges in orthogonality regularization.

## 1 Introduction

Despite the widespread adoption and success of deep convolutional neural networks in various applications Krizhevsky et al. (2012); Simonyan & Zisserman (2015); He et al. (2016); Tan & Le (2019); Ding et al. (2022), they are not without challenges. Issues such as vanishing gradient Bengio et al. (1994); Glorot & Bengio (2010), feature statistic shifts Ioffe & Szegedy (2015), and saddle points Dauphin et al. (2014) hinder the training of CNNs. To mitigate these challenges, researchers have explored a range of techniques. These include methods for better parameter initialization Saxe et al. (2014), normalization techniques to stabilize internal activations Ioffe & Szegedy (2015), and architectures that support residual learning Srivastava et al. (2015); He et al. (2015).

Additionally, there's a growing interest in methods that focus on the relationships between convolutional filters. Correlation regularization aims to align the correlation information of convolutional filters with a predefined ideal distribution Liu et al. (2020); Wang et al. (2021). In contrast, orthogonality regularization seeks to ensure that processed convolutional filter information is orthogonal. A notable subtype of this is the kernel orthogonality regularization, enforces the orthogonality of the kernel function of the convolutional filters Ozay & Okatani (2016); Harandi & Fernando (2016).

In contrast to the prevailing focus on optimizing measure of the distance between the kernel and the identity matrix Xie et al. (2017); Bansal et al. (2018) and the influence of the isometry property on

training Qi et al. (2020); Huang et al. (2018), our work identifies a subtle but significant challenge in orthogonality regularization. Specifically, when there's an emphasis on minimizing task loss, the optimization objective for orthogonality regularization can become challenging to achieve. This complexity often results in a gap, causing traditional orthogonality regularization to underperform.

To tackle this challenge, we introduce disentangled orthogonality regularization. This method combines the advantages of correlation and orthogonality regularization, aiming to push the distribution of convolutional filters towards orthogonality stably without compromising the constraint on filter norms. Our extensive experiments show that this approach surpasses existing methods in achieving near-orthogonality. While models with shallow architectures benefit from this method, the advantage wanes as the network depth increases. This observation prompts a pivotal question:

<p align="center">Is the pursuit of strict orthogonality always justified?</p>

In response to this, our findings indicate that while better near-orthogonality should correlates with enhanced performance, the benefits plateau in deeper models. This calls for a reevaluation of the existing strict orthogonality paradigm, especially for deep architectures.

Building on these insights, we introduce a relaxation theory. Following its principles, we craft a variant of the disentangled orthogonality regularization that offers a more flexible approach to orthogonality in deep networks. This variant specifically addresses certain convolutional filters that were previously excluded from the kernel orthogonality definition. Additionally, with the introduction of the concept of transition dimension, filters that were previously included in orthogonality regularization can now be better optimized, unlocking their full potential. This refined approach effectively overcomes the challenges associated with strict orthogonality, particularly in the background space of deeper networks, culminating in improved model performance.

## 2 RELATED WORKS

The benefits of orthogonality filters were first researched in recurrent neural networks (RNNs) to address gradient vanishing or exploding problems Arjovsky et al. (2016); Wisdom et al. (2016); Dorobantu et al. (2016). In the context of RNNs, Casado & Martínez-Rubio (2019) introduced parameterization from exponential maps to achieve computational efficiency. The trade-offs between soft and hard orthogonality in RNNs are further elaborated upon in Vorontsov et al. (2017).

Moving to convolutional neural networks (CNNs), the role of orthogonality in stabilizing training has been highlighted in Rodríguez et al. (2017); Bansal et al. (2018); Xie et al. (2017). Methods to preserve the orthogonality property during CNN training are investigated in Harandi & Fernando (2016); Ozay & Okatani (2016), with a focus on Stiefel manifold-based optimization techniques. In practice, orthogonality regularization has been found to enhance the training of image generation models Brock et al. (2019; 2017); Miyato et al. (2018); Shukla et al. (2019); Peebles et al. (2020).

A particular focus in the literature has been on the imposition of orthogonality on the convolutional filters of networks, with empirical results validating this approach Harandi & Fernando (2016); Ozay & Okatani (2016); Xie et al. (2017); Qi et al. (2020). When it comes to quantifying orthogonality, a significant portion of research has centered around the residual between the Gram matrix and the identity matrix. The introduction of Frobenius norm on orthogonality regularization has been a popular method in previous works, aiming to optimize the Gram matrix by minimizing the Frobenius norm between the identity matrix and the Gram matrix Xie et al. (2017); Huang et al. (2018). An improvement to this method was introduced by (Kim & Yun, 2022), which provides a more balanced approach for handling layers with different filter numbers. On the other hand, the Spectral Restricted Isometry Property Regularization (SRIP) replaces the Frobenius norm with the spectral norm, enhancing the network's generalization ability and achieving state-of-the-art performance Bansal et al. (2018). Furthermore, (Wang et al., 2020) proposes alternative approaches working on improving the kernel function, exploring approximations in spaces with desirable properties.

In the domain of correlation regularization, a common strategy involves guiding the distribution of filters towards a predefined ideal. While (Liu et al., 2020) drew inspiration from the Thomson Problem, (Wang et al., 2021) aimed to approximate the solution of the Tammes problem. However, typical correlation regularization techniques derive from the ideal that filters are situated within a unit sphere, often neglecting the impact of filter norms on training.

## 3 DISENTANGLED ORTHOGONALITY AND RELAXATION THEORY

### 3.1 PRELIMINARY

First, we establish a unified notation, clarify its relevance, and define the terminology in the context:

- Convolutional filters: transformation matrix $\boldsymbol{K}$ of convolutional filters, $\boldsymbol{K} \in \mathbb{R}^{o \times i \times k_h \times k_w}$, where:
  - $o$: Number of output channels.
  - $i$: Number of input channels.
  - $k_h$: Height of the convolutional kernel.
  - $k_w$: Width of the convolutional kernel.

  The convolutional filters $\boldsymbol{K}$ can be reshaped to $\mathbb{R}^{o \times (i \times k_h \times k_w)}$, representing $o$ convolutional filters:

  $$\boldsymbol{K} = \begin{pmatrix} \text{—} & \boldsymbol{k}_1 & \text{—} \\ & \vdots & \\ \text{—} & \boldsymbol{k}_o & \text{—} \end{pmatrix}, \text{where } \boldsymbol{k}_i \text{ induce a linear map } \langle \boldsymbol{k}_i, \cdot \rangle : \mathbb{R}^{i \times k_h \times k_w} \mapsto \mathbb{R} \quad (1)$$

  Convolutional filters $\boldsymbol{K}$ maps the stacked input patches to the output space with dimension $\mathbb{R}^o$. The normalized filters, denoted by $\tilde{\boldsymbol{k}}_i = \frac{\boldsymbol{k}_i}{\|\boldsymbol{k}_i\|}$, form the normalized convolutional filters $\tilde{\boldsymbol{K}}$

- Kernel matrix / Gram matrix: the kernel function of convolutional filters $\boldsymbol{K}$, denoted as $\boldsymbol{K}\boldsymbol{K}^\top$, whose entries are given by the inner product $\text{Gram}_{ij} = \langle \boldsymbol{k}_i, \boldsymbol{k}_j \rangle$. The term orthogonality for the convolutional filters $\boldsymbol{K}$ specifically refers to the condition where the Gram matrix is equal to the identity matrix, i.e., $\boldsymbol{K}\boldsymbol{K}^\top = \boldsymbol{I}_{o \times o}$. In prior research, strict orthogonality regularization is characterized by an optimization on $\boldsymbol{K}$ to minimize the measure of kernel residual:

  $$\|\boldsymbol{K}\boldsymbol{K}^\top - \boldsymbol{I}_{o \times o}\| \quad (2)$$

  with better strict orthogonality implying that the measure on residual $\|\boldsymbol{K}\boldsymbol{K}^\top - \boldsymbol{I}_{o \times o}\|$ is smaller.

- Over-determined / Less-determined: these terms describe the relationship between the rows and columns in the reshaped convolutional filters $\boldsymbol{K}$. Convolutional filters $\boldsymbol{K}$ with $o \leq (i \times k_h \times k_w)$ is defined as less-determined. In Appendix A.3, we discuss over-determined convolutional filters $\boldsymbol{K}$ is theoretically inaccessible to strict orthogonality: $\boldsymbol{K}\boldsymbol{K}^\top \neq \boldsymbol{I}_{o \times o}, o > (i \times k_h \times k_w)$

### 3.2 STRICT KERNEL ORTHOGONALITY REGULARIZATION, DISENTANGLED NORM

This section elaborates on the concept of the disentangled norm and its derivation. This norm aims to improve upon traditional orthogonality regularization methods by providing a more nuanced measure of orthogonality that accounts for both off-diagonal and diagonal properties.

Before exploring the disentangled norm, let's grasp the motivation behind strict orthogonality regularization. Kernel strict orthogonality regularization tends to push the Gram matrix $\boldsymbol{K}\boldsymbol{K}^\top$ towards the identity matrix $\boldsymbol{I}$. This regularization results in two main effects:

- On diagonal entry: Strict orthogonality aims for the diagonal of the Gram matrix to have unit values, indicating filters with unit norms:

  $$\text{diag}(\boldsymbol{K}\boldsymbol{K}^\top)_i = \text{Gram}_{ii} = \langle \boldsymbol{k}_i, \boldsymbol{k}_i \rangle \to 1 \quad (3)$$

  Having filters with unit norms is desirable as it ensures that each filter contributes effectively during the convolution operation. In practice, orthogonality regularization on the diagonal constrains the variance of the convolutional filters based on their norms. This constraint guarantees that all filters, even those with smaller norms, contribute effectively, preventing overly weak output features.

- On off-diagonal entry: Strict orthogonality enforces off-diagonal entries towards zero:

  $$\text{Gram}_{ij} = \langle \boldsymbol{k}_i, \boldsymbol{k}_j \rangle \to 0 \quad (4)$$

  In practice, this results in reduced correlation between convolutional filters, assisting the network in minimizing filter redundancy.

With these insights, we now turn to the derivation of the disentangled norm, highlighting its advantages in strict kernel orthogonality optimization.

We now disentangle the diagonal and off-diagonal components from the Gram matrix $\boldsymbol{K}\boldsymbol{K}^\top$. Since $\boldsymbol{K}\boldsymbol{K}^\top$ is a real symmetric matrix, it suffices to consider its lower triangular and the diagonal:

$$\text{LowerTriangular}(\boldsymbol{K}\boldsymbol{K}^\top) = \begin{bmatrix} 0 & 0 & \cdots & 0 \\ \langle \boldsymbol{k}_2, \boldsymbol{k}_1 \rangle & 0 & \cdots & 0 \\ \vdots & \vdots & \ddots & \vdots \\ \langle \boldsymbol{k}_n, \boldsymbol{k}_1 \rangle & \langle \boldsymbol{k}_n, \boldsymbol{k}_2 \rangle & \cdots & 0 \end{bmatrix}, \text{diag}(\boldsymbol{K}\boldsymbol{K}^\top) = \begin{bmatrix} \langle \boldsymbol{k}_1, \boldsymbol{k}_1 \rangle \\ \vdots \\ \langle \boldsymbol{k}_n, \boldsymbol{k}_n \rangle \end{bmatrix} \tag{5}$$

Considering that the correlation between two filter $\text{Corr}(\boldsymbol{k}_i, \boldsymbol{k}_j) = \frac{\langle \boldsymbol{k}_i, \boldsymbol{k}_j \rangle}{\|\boldsymbol{k}_i\|\|\boldsymbol{k}_j\|}$ provides a measure of orthogonality between them regardless of the influence from their norms. Furthermore, the lower triangular of correlation matrix aligns with the lower triangular of normalized Gram matrix $\tilde{\boldsymbol{K}}\tilde{\boldsymbol{K}}^\top$. In computation of the disentangled norm of the kernel residual:

$$\|\boldsymbol{K}\boldsymbol{K}^\top - \boldsymbol{I}_{o\times o}\| = \left\|\text{LowerTriangular}(\tilde{\boldsymbol{K}}\tilde{\boldsymbol{K}}^\top) - \boldsymbol{0}_{o\times o}\right\|_F + \lambda \left\|\text{diag}(\boldsymbol{K}\boldsymbol{K}^\top) - \boldsymbol{1}_{o\times 1}\right\|_F \tag{6}$$

$\|\cdot\|_F$ refer to the Frobenius norm, $\lambda$ is a balance coefficent between disentangled correlation loss and diagonal loss. In Appendix A.2, we discuss the motivation of the disentangled norm.

For the sake of computational efficiency, we compute the diagonal of the Gram matrix $\text{diag}(\boldsymbol{K}\boldsymbol{K}^\top)$ from the filter norms of $\boldsymbol{K}$. With these filter norms, we can derive the normalized matrix $\tilde{\boldsymbol{K}}$. This allows us to obtain the lower triangular of the normalized Gram matrix $\text{LowerTriangular}(\tilde{\boldsymbol{K}}\tilde{\boldsymbol{K}}^\top)$.

### 3.3 Relaxation theory, relaxed Kernel disentangled orthogonality

#### 3.3.1 Relaxation on over-determined layers

In this section, we propose the relaxation theory on kernel orthogonality regularization. As highlighted in Appendix A.3, over-determined $\boldsymbol{K}$ inherently do not comply with the definition of strict kernel orthogonality. Given a matrix $\boldsymbol{K}_{o\times n}$ with rows $\boldsymbol{k}_1, \boldsymbol{k}_2, \ldots, \boldsymbol{k}_o$, where $n = i \times k_h \times k_w$ and $o > n$, it can be asserted that there always exist filter pairs $(\boldsymbol{k}_i, \boldsymbol{k}_j)$ in $\boldsymbol{K}$ such that their correlation $\text{Corr}(\boldsymbol{k}_i, \boldsymbol{k}_j) \neq 0$. It is a natural approach to consider relaxing these non-zero correlation pairs from the orthogonality regularization. Building on this idea, this observation leads to a pivotal question:

How can we identify these pairs in the correlation matrix that should be relaxed?

Directly determining the exact number of pairs that should be relaxed is challenging. However, by analyzing the components in the lower triangular of the correlation matrix, we can obtain an approximation of the lowerbound number for the numbers of the pairs that should be relaxed. We denote the vectorized LowerTriangular of the correlation matrix as $\boldsymbol{c}$, with the dimension $\frac{o\times(o-1)}{2}$:

$$\boldsymbol{c} = \text{vec}(\text{LowerTriangular}(\tilde{\boldsymbol{K}}\tilde{\boldsymbol{K}}^\top)), \boldsymbol{c}_i \in [-1, 1] \tag{7}$$

For the overdetermined $\boldsymbol{K}_{o\times n}$, the convolutional filter can form $\text{span}\{\boldsymbol{k}_1, \ldots, \boldsymbol{k}_o\}$ with a maximum rank $n = \min(o, n)$. As shown in Fig. 1, under this constraint on the rank, we construct an orthogonal structure with $n$ filters, these filters, which support the rank, are termed as structural filters. We assign these $n$ structural filters, to the label $\ell(\boldsymbol{k}_i)$ from the index $\{1, \ldots, n\}$. The rest $o - n$ filters, which are supposed to adjust the relative distribution of filters for an enhanced representation, are termed as flexible filters. In addition to classify the filters into two types, we can assign the nearest structural filter index to the flexible filters in correlation measure and the structural filter index assigned to itself:

$$\boldsymbol{1}_{\text{structural}}(\boldsymbol{k}_i) = \begin{cases} 1 & \text{structural filter } \boldsymbol{k}_i \\ 0 & \text{flexible filter } \boldsymbol{k}_i \end{cases}, \ell(\boldsymbol{k}_i) = j = \operatorname*{arg\,max}_{\boldsymbol{1}_{\text{structural}}(\boldsymbol{k}_j)=1} |\text{Corr}(\boldsymbol{k}_i, \boldsymbol{k}_j)| \tag{8}$$

In accordance with the equality of $(\boldsymbol{1}_{\text{structural}}(\boldsymbol{k}_i), \boldsymbol{1}_{\text{structural}}(\boldsymbol{k}_j))$, $(\ell(\boldsymbol{k}_i), \ell(\boldsymbol{k}_j))$, we can categorize the entries $\boldsymbol{c}_{\boldsymbol{k}_i, \boldsymbol{k}_j}$ in $\boldsymbol{c}$ into four types:

- $\boldsymbol{1}_{\text{structural}}(\boldsymbol{k}_i) = \boldsymbol{1}_{\text{structural}}(\boldsymbol{k}_j) = 1$: $\boldsymbol{c}_{(1,1)}$ remain subjected to orthogonality regularization.

- $\mathbf{1}_{\text{structural}}(\boldsymbol{k}_i) = \mathbf{1}_{\text{structural}}(\boldsymbol{k}_j) = 0$: $\boldsymbol{c}_{(1,0)}$ with a relative small magnitude, since the flexible filters will be redistributed to alleviate filter redundancy in training to get the better representation.
- $\mathbf{1}_{\text{structural}}(\boldsymbol{k}_i) \neq \mathbf{1}_{\text{structural}}(\boldsymbol{k}_j)$: We can further categorize these type $\boldsymbol{c}_{\boldsymbol{k}_i, \boldsymbol{k}_j}$
  - $\ell(\boldsymbol{k}_i) = \ell(\boldsymbol{k}_j)$: $\boldsymbol{c}_{(0,1)}$ share the same label, thus tend to have large correlation.
  - $\ell(\boldsymbol{k}_i) \neq \ell(\boldsymbol{k}_j)$: $\boldsymbol{c}_{(0,0)}$ have different label, the flexible filter $\boldsymbol{k}_i$ may locate in the orthogonal complement of the structural filter $\boldsymbol{k}_j^\perp$, as shown in the first segment of Fig. 1, $\boldsymbol{k}_{65} \in \boldsymbol{k}_2^\perp$

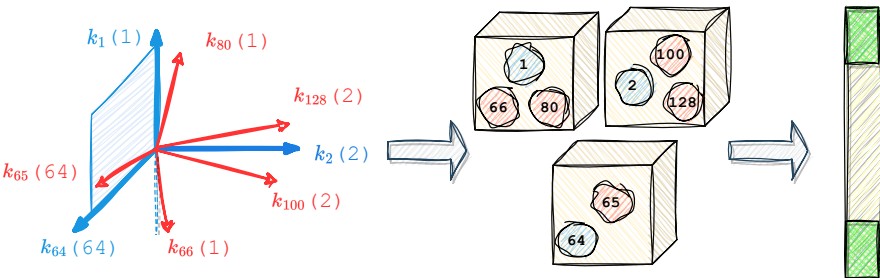

Figure 1: llustration of the relaxed orthogonality regularization on over-determined layers. The first segment shows the construction of the structural filters and flexible filters. The second segment shows the assignment of flexible filters to the respective structural filters. The third segement shows the relaxed $\boldsymbol{c}$ in yellow color under strict orthogonality regularization

Among $\boldsymbol{c}_{(1,0)}, \boldsymbol{c}_{(0,1)}, \boldsymbol{c}_{(0,0)}$, it is observed that $\boldsymbol{c}_{(0,1)}$ is the category that should primarily be relaxed. To investigate the $\text{card}(\boldsymbol{c}_{(0,1)})$, $\text{card}(\cdot)$ stands for the number of elements in a set. $\text{card}(\boldsymbol{c}_{(0,1)})$ can be split to a sum of pairs in $\boldsymbol{c}$ based on all structural filter with the same label:

$$\text{card}(\boldsymbol{c}_{(0,1)}) = \Sigma_{\mathbf{1}_{\text{structural}}(\boldsymbol{k}_i)=1} \text{card}(\{(\boldsymbol{k}_i, \boldsymbol{k}_j)|\ell(\boldsymbol{k}_i) = \ell(\boldsymbol{k}_j), \boldsymbol{k}_i \neq \boldsymbol{k}_j\}) \quad (9)$$

Different filter distributions can lead to variations in $\text{card}(\boldsymbol{c}_{(0,1)})$, we choose to give an approximation to $\text{card}(\boldsymbol{c}_{(0,1)})$ by Monte Carlo method. For ease of research, we assume that the label assignment of flexible filter to structural filter follow a uniform distribution, we can approximate the shared-label correlation pair in $\boldsymbol{c}$, fixed structural filter in $m$ times Monte Carlo, as depicted in the second segment in Fig. 1. $U_m(n)$ acts as the uniform distribution in $m$ experiment.

$$\tilde{\text{card}}(\boldsymbol{c}_{(0,1)}) = \frac{1}{m}\Sigma_m \text{card}(\boldsymbol{c}_{m(0,1)}), \forall \boldsymbol{k}_i \in \boldsymbol{K}, \mathbf{1}_{\text{structural}}(\boldsymbol{k}_i) = 0, l_m(\boldsymbol{k}_i) \sim U_m(n) \quad (10)$$

$\tilde{\text{card}}(\boldsymbol{c}_{(0,1)})$ serve as an approximation to the lower bound of the should-be-relaxed pairs in $\boldsymbol{c}$:

$$\tilde{\text{card}}(\boldsymbol{c}_{(0,1)}) \to \text{card}(\boldsymbol{c}_{(0,1)}) \leq \text{card}(\boldsymbol{c}_{(0,1)}) + \text{card}(\boldsymbol{c}_{(0,0)}) + \text{card}(\boldsymbol{c}_{(1,0)}) \quad (11)$$

We split $\boldsymbol{c}$ into the positive part and the negative part $\boldsymbol{c} = \boldsymbol{c}^+ - \boldsymbol{c}^-$. As the positive correlation and negative correlation are symmetric in representation Kuo (2016), let $\text{topk}_k(\cdot)$ represent the function that extracts the $k$-largest elements from a vector. We respectively remove $\text{topk}_k(\boldsymbol{c}^+)$ and $\text{topk}_k(\boldsymbol{c}^-)$, $k = \tilde{\text{card}}(\boldsymbol{c}_{(0,1)})/2$ from the vector $\boldsymbol{c}^+$ and $\boldsymbol{c}^-$. As shown in the third segment in Fig. 1, the relaxed disentangled orthogonality regularization on over-determined $\boldsymbol{K}$:

$$\|\boldsymbol{K}\boldsymbol{K}^\top - \boldsymbol{I}_{o \times o}\| = \|\boldsymbol{c}^+ \backslash \text{topk}_k(\boldsymbol{c}^+) - \mathbf{0}\|_F + \|\boldsymbol{c}^- \backslash \text{topk}_k(\boldsymbol{c}^-) - \mathbf{0}\|_F \quad (12)$$

For the overdetermined filters $\boldsymbol{K}$, we enforce the orthogonality on their correlation and remove the constraint on their norm, the motivation for this discussed in Appendix A.4.

### 3.3.2 RELAXATION ON LESS-DETERMINED LAYERS

In this section, we introduce a method for relaxation on less-determined layers, to address the declined performance enhancement observed in deeper models equipped with strict disentangled orthogonality regularization. Contrary to the focus on convolutional filters $\boldsymbol{K}$ for relaxation on the over-determined layers, we shift our attention to the data representation $\boldsymbol{X}$ in the relaxation on less-determined layers.

Consider the expressiveness of a transform in a hidden layer given by: $\boldsymbol{X}_j = \boldsymbol{K}_j(\boldsymbol{X}_{j-1})$, where $\boldsymbol{X}_j$ and $\boldsymbol{X}_{j-1}$ are the representations at layers $j$ and $j - 1$:

- Less expressive: given rank inequality: $\text{rank}(\boldsymbol{AB}) \leq \min(\text{rank}(\boldsymbol{A}), \text{rank}(\boldsymbol{B}))$, the rank of the output representation $\boldsymbol{X}_j$ is constrained as:

$$\text{rank}(\boldsymbol{X}_j) \leq \min(\text{rank}(\boldsymbol{K}_j), \text{rank}(\boldsymbol{X}_{j-1})) \tag{13}$$

  If $\text{rank}(\boldsymbol{K}_j)$ is too small, it can lead to a collapse in the rank of $\boldsymbol{X}_j$, reducing the expressiveness of the representation. This serves as a primary motivation for orthogonality regularization.

- Overly expressive: if $\boldsymbol{K}_j$ is overly expressive, there's a risk of overfitting to the training data. Consider splitting $\boldsymbol{K}_j$ as: $\begin{bmatrix} \boldsymbol{K}^* \\ \boldsymbol{K} \backslash \boldsymbol{K}^* \end{bmatrix}$, where $\boldsymbol{K}^*$ contains filters producing meaningful features, while $\boldsymbol{K} \backslash \boldsymbol{K}^*$ contains filters that optimize training loss but might not generalize well. To mitigate the effects of non-contributive filters in $\boldsymbol{K}_j$, regularization techniques like dropout Zagoruyko & Komodakis (2016) can be employed.

From our analysis, we deduce that there should be an optimal $\text{rank}^*(\boldsymbol{K}_j)$, which we denote as transition dimension. The challenge then becomes finding an approximation to $\text{rank}^*(\boldsymbol{K}_j)$. Instead of imposing strict orthogonality regularization on $\mathbb{R}^{o_j}$, we focus on the strict orthogonality on the transition dimension $\text{rank}^*(\boldsymbol{K}_j)$. This approach is referred to as relaxed disentangled orthogonality regularization on less-determined layers $\boldsymbol{K}_j \in \mathbb{R}^{o_j \times (i_j \times (k_h)_j, (k_w)_j)}$.

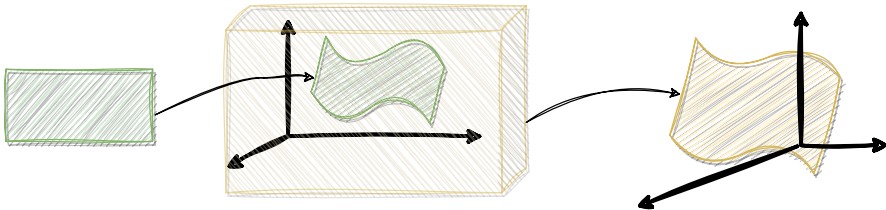

Figure 2: Illustration of the concept of the transition dimension. Moving from left to right in the illustration, The intrinsic dimension of data representation $\boldsymbol{X}_{j-1}$ is represented in green plane. The transition dimension, which formed by $\boldsymbol{K}_j^*(\boldsymbol{X}_{j-1})$, $\boldsymbol{K}_j^*$ with $\text{rank}(\boldsymbol{K}_j^*)$. The output dimension is denoted by $\mathbb{R}^{o_j}$, within which the transition dimension embedded in as a low-rank manifold. The transition dimension can align with specific $\mathbb{R}^{o_j}$ with a relative low dimension $o_j$.

Consider the transform $\boldsymbol{X}_j = \boldsymbol{K}_j(\boldsymbol{X}_{j-1})$ with transition dimension shown in Fig. 2, To approximate the transition dimension, we consider three attributes from the dataset and filters in architectures:

- Intrinsic dimension IntDim: $\boldsymbol{K}_j$ should avoid less-expressive from Equation (13), we take the intrinsic dimension of dataset IntDim, acting as an approximation to $\min(\text{rank}(\boldsymbol{X}))$. $K_j^*$ should satisfy $\text{rank}(\boldsymbol{K}_j^*) \geq \min(\text{rank}(\boldsymbol{X}))$.

- Dataset attribute $n$: For an infinite-depth network operating on a dataset with $n$ labels, the network can learn a complex map that disentangle the features into $n$ direction. Thus $K_j^*$ should satisfy $\text{rank}(\boldsymbol{K}_j^*) \geq n$ to disentangle features.

- Layerwise capacity LC: The transition dimension should vary depending on the filters width. This refers to the following condition under the same model: $(\boldsymbol{K}_1)_{o_{j_1} \times \cdot}, (\boldsymbol{K}_2)_{o_{j_2} \times \cdot}, o_{j_1} \geq o_{j_2} \Rightarrow \text{rank}(\boldsymbol{K}_1^*) \geq \text{rank}(\boldsymbol{K}_2^*)$, While this layerwise capacity ascent should not be linearly proportional to the output-channel dimension ascent. We consider the logarithm(10) based on the narrowest filter width, to corporate with the IntDim from dataset representation: $\text{LC} = \text{IntDim} \times \log_{10}(o_j)$

Based on the discussion above, there comes to the approximation of the $\text{rank}(\boldsymbol{K}_j^*)$:

$$\text{rank}(\boldsymbol{K}_j^*) \leftarrow \tilde{\text{rank}}(\boldsymbol{K}_j^*) = \min\left[\max\left(\text{IntDim}, n\right), \text{LC}\right] \tag{14}$$

With $\tilde{\text{rank}}(\boldsymbol{K}_j^*)$, we propose the relaxed orthogonality on the specific less-determined layers $\boldsymbol{K}_{o \times n}, o > \tilde{\text{rank}}(\boldsymbol{K}_j^*)$. The computation of cardinality $k$ (Equation (10)) replace $n$ with $\tilde{\text{rank}}(\boldsymbol{K}_j^*)$:

$$\|\boldsymbol{KK}^\top - \boldsymbol{I}_{o \times o}\| = \|\boldsymbol{c}^+ \backslash \text{topk}_k(\boldsymbol{c}^+) - \boldsymbol{0}\|_F + \|\boldsymbol{c}^- \backslash \text{topk}_k(\boldsymbol{c}^-) - \boldsymbol{0}\|_F + \lambda \left\|\text{diag}(\boldsymbol{KK}^\top) - \boldsymbol{1}\right\|_F \tag{15}$$

## 4 EXPERIMENTS

Our experiments were conducted on the CIFAR100 and CIFAR10 datasets Krizhevsky (2009). Each dataset consists of 60,000 images with dimensions $32 \times 32$. CIFAR100 has 100 unique labels, while CIFAR10 features 10. Adhering to the partitioning method from He et al. (2016), we designated 45,000 images for training and 5,000 for validation from the 50,000 training images. The remaining 10,000 images were used as our test set. In the data preprocessing phase, we applied a random crop transformation with a 4-pixel padding and a subsequent random horizontal flip to the $32 \times 32$ input images for augmentation. These images were then normalized using mean and standard deviation values computed directly from the dataset.

Our choice of models included several classical ResNet architectures He et al. (2016), which were selected due to their typical and lightweight nature. This included narrow channel variants like ResNet20, ResNet32, and ResNet56, as well as broader channel variants such as ResNet18 and ResNet34. Additionally, we explored the ResNet50 model, which features a bottleneck structure, and the WideResNet $28 \times 10$ architecture.

For the training configuration, we adhered to the methodology described in He et al. (2016). For CIFAR10, we employed the SGD optimizer with a Nesterov Momentum of 0.9, training over 160 epochs. We initiated the learning rate at 0.1, decreasing it by a factor of 10 post the 80th and 120th epochs using MultiStepLR. For CIFAR100, we trained over 200 epochs with the SGD optimizer and a Nesterov Momentum of 0.9. The learning rate was initially set at 0.1 and adjusted by a factor of 5 post the 80th, 120th, and 160th epochs. We maintained a batch size of 128 for the SGD optimizer and deployed two RTX 4090 GPUs for our experiments.

### 4.1 HYPERPARAMETER SCHEME

In this section, we present our scheme for hyperparameter tuning:

- Balance between Task Loss and Regularization Loss:

  Let $L_{\text{task}}$ denote the task-specific loss (e.g., classification loss) and $L_{\text{reg}}$ represent the orthogonality regularization. As discussed in Appendix A.2, The proportion of each loss type to the overall loss, $L_{\text{total}}$, is crucial. It's imperative to ensure:

  $$\frac{L_{\text{task}}}{L_{\text{total}}} \geq \theta$$

  where $\theta$ is a threshold ensuring the significance of the task-specific loss.

  To initialize the hyperparameter for orthogonality regularization, we adhere to the following criterion:

  $$\left| \frac{\sum L_{\text{reg}}}{L_{\text{total}}} - \beta_{\text{reg}} \right| \leq \epsilon_{\text{reg}}$$

  where $\beta_{\text{reg}}$ is the desired balance for regularization, set to 10%, and $\epsilon_{\text{reg}}$ is a tolerance level set to 1%.

  During subsequent training, in reference to the Scheme Change for Regularization Coefficients Bansal et al. (2018), we adjust our method at certain epochs. These include the beginning and midpoint of the second and third learning stages, as well as the start of the fourth learning stage. If the sum of the ratios of the regularization terms exceeds the set ratio, it is scaled down to less than 40%. This strategy ensures a balanced and flexible approach to model training, adjusting the contribution of different loss components as training progresses.

- Balance between Diagonal Loss and Correlation Loss: When balancing diagonal loss and correlation loss, we prioritize correlation loss as it plays a more critical role in orthogonality regularization. Similar to the task balance control, at the same epoch where we monitor the balance between task-specific loss and regularization loss, we set the balance between diagonal loss and correlation loss to be 10% and $\epsilon_{\text{disentangled}}$ to 5%.

- Relaxation of Transition Dimension: For the approximation of transition dimension in Equation (14) the Layerwise Capacity LC and dataset attributes $n$ can directly be obtained from the model and the dataset. For the intrinsic dimension, we refer to the research of Pope et al. (2021), setting the intrinsic dimension $\text{IntDim} = 30$ for CIFAR10 and CIFAR100 in the experiments.

## 4.2 ON THE PERFORMANCE GAINS UNDER ORTHOGONALITY REGULARIZATION

In the subsequent sections, we systematically examine the impact of orthogonality regularization. Relaxed disentangled orthogonality, as defined in Section 3.3.1, is applied by default to the over-determined layers under both strict and relaxed disentangled orthogonality. We set the default transition dimension for over-determined convolutional filters as $\tilde{\mathrm{rank}}(\boldsymbol{K}^*_{o \times n}) = n$.

Upon analyzing narrow ResNet models, we found that strict orthogonality could might hinder performance in shallow variants, which are the models with the lowest output dimension in our experiment. Due to the low-rank nature of this type of ResNet, the strict orthogonality regularization may lead the convolutional filters to be over-regularized. However, the introduction of the transition dimension in narrow width ResNet models can still enhance their performance. As the network depth increased, the advantage of strict orthogonality regularization on the output dimension became evident. Moreover, relaxed orthogonality regularization on the transition dimension led to further performance improvement.

In the case of mid-width ResNet models, an increased output dimension, resulting from the convolutional filters with more filters, leads to a more complex data representation. Baseline regularization methods like Frobenius Xie et al. (2017) and SRIP Bansal et al. (2018) improved model performance, and the application of relaxed orthogonality showed its advantage in shallow models like ResNet. In comparison, the overdetermined layer in the bottleneck structure seemed to challenge the effectiveness of strict orthogonality regularization methods. However, our proposed relaxation on overdetermined layers not only stabilized the training process but also led to superior performance in ResNet models with bottleneck structures.

For WideResNet models, the advantages of both baseline strict orthogonality regularization and relaxed orthogonality on the transition dimension were evident. However, strict disentangled orthogonality regularization on the output dimension appeared to be the least effective for performance improvement. This might be due to the fact that WideResNet models have the highest output dimension in our experiment, making the introduction of the transition dimension in WideResNet models very significant.

Table 1: The table showcases test accuracy outcomes for various scenarios, represented as mean and standard deviation values from three runs with random seeds. Different orthogonality regularization methods are listed along the rows. The term Vanilla refers to optimization without regularization, Strict indicates strict disentangled orthogonality regularization in the output space, and Relaxed represents relaxed disentangled orthogonality in the transition dimension estimated by Equation (14). WRN 28×10 in the last row represents WideResNet 28×10.

| Test Acc Mean/Std | Vanilla | Frobenius | SRIP | Strict | Relaxed |
|---|---|---|---|---|---|
| | | | 16-32-64 | | |
| ResNet20 | $91.65 \pm 0.15$ | $91.68 \pm 0.11$ | $91.75 \pm 0.15$ | $91.59 \pm 0.13$ | $\mathbf{91.90 \pm 0.12}$ |
| ResNet32 | $92.81 \pm 0.21$ | $92.81 \pm 0.12$ | $92.85 \pm 0.14$ | $92.73 \pm 0.19$ | $\mathbf{93.04 \pm 0.18}$ |
| ResNet56 | $93.25 \pm 0.17$ | $93.30 \pm 0.16$ | $93.47 \pm 0.15$ | $93.17 \pm 0.20$ | $\mathbf{93.53 \pm 0.09}$ |
| | | | 64-128-256-512 | | |
| ResNet18 | $76.51 \pm 0.18$ | $76.87 \pm 0.13$ | $77.10 \pm 0.18$ | $77.09 \pm 0.17$ | $\mathbf{77.35 \pm 0.11}$ |
| ResNet34 | $77.08 \pm 0.22$ | $77.43 \pm 0.16$ | $77.69 \pm 0.12$ | $77.63 \pm 0.19$ | $\mathbf{77.85 \pm 0.17}$ |
| ResNet50 | $77.43 \pm 0.16$ | $77.82 \pm 0.17$ | $77.71 \pm 0.22$ | $78.12 \pm 0.13$ | $\mathbf{78.50 \pm 0.16}$ |
| | | | 160-320-640 | | |
| WRN 28×10 | $79.32 \pm 0.16$ | $79.82 \pm 0.13$ | $80.11 \pm 0.12$ | $79.73 \pm 0.22$ | $\mathbf{80.23 \pm 0.12}$ |

## 4.3 ON THE NEAR-ORTHOGONALITY UNDER ORTHOGONALITY REGULARIZATION

In this section, we will examine the extent of near-orthogonality under various orthogonality regularizations. We will focus on the following models: narrow variants of ResNet (ResNet56), mid-width variants of ResNet (ResNet18), and the wider variant WideResNet (WRN28×10). For the less-determined layers, we exhibit the average statistics of all layers in the same output dimension.

Notably, no regularization scheme can achieve perfect orthogonality in the less-determined layers of the well-trained models. Starting with the narrowest ResNet variant, ResNet56, we observe that due to its low-dimension output space, the well-trained model under strict disentangled orthogonality almost

Table 2: In the table, we quantify the near-orthogonality of a specific layer by analyzing the statistics of the correlation matrix and the diagonal. The mean of the lower triangular part of the correlation matrix represents the average degree to which filters in a specific transformation approach zero-correlation. Furthermore, the standard deviation of the correlation indicates the stability of near-orthogonality. Values separated by '/' represent the mean and the average diagonal of the layer

| ResNet56 | Layer3 Downsample | Layer1 [16,144] | Layer2 [32,288] | Layer3 [64,576] |
|---|---|---|---|---|
| Vanilla | $0.01 \pm 0.10/0.03$ | $0.04 \pm 0.25/0.06$ | $0.01 \pm 0.11/0.05$ | $0.01 \pm 0.06/0.18$ |
| Frobenius | $0.02 \pm 0.19/0.13$ | $-0.00 \pm 0.05/0.27$ | $-0.00 \pm 0.09/0.22$ | $-0.01 \pm 0.09/0.42$ |
| SRIP | $0.01 \pm 0.18/0.17$ | $0.00 \pm 0.02/0.23$ | $0.00 \pm 0.09/0.24$ | $-0.01 \pm 0.09/0.45$ |
| Strict | $0.01 \pm 0.13/0.17$ | $\mathbf{0.00 \pm 0.00/0.95}$ | $\mathbf{-0.00 \pm 0.01/0.90}$ | $\mathbf{0.00 \pm 0.01/1.12}$ |
| Relaxed | $\mathbf{0.00 \pm 0.13/0.20}$ | $0.00 \pm 0.01/0.31$ | $-0.00 \pm 0.02/0.30$ | $-0.00 \pm 0.03/0.60$ |
| ResNet18 | Layer3 Downsample | Layer2 [128,1152] | Layer3 [256,2304] | Layer4 [512,4608] |
| Vanilla | $0.01 \pm 0.10/0.03$ | $0.04 \pm 0.25/0.06$ | $0.01 \pm 0.11/0.05$ | $0.01 \pm 0.06/0.16$ |
| Strict | $\mathbf{0.01 \pm 0.10/0.11}$ | $\mathbf{0.00 \pm 0.01/0.26}$ | $\mathbf{0.00 \pm 0.02/0.16}$ | $\mathbf{0.01 \pm 0.02/0.18}$ |
| WRN 28 × 10 | Layer3 Downsample | Layer1 [160,1440] | Layer2 [320,2880] | Layer3 [640,5760] |
| Vanilla | $0.01 \pm 0.10/0.03$ | $0.04 \pm 0.25/0.06$ | $0.01 \pm 0.11/0.05$ | $0.01 \pm 0.06/0.18$ |
| Strict | $\mathbf{0.01 \pm 0.05/0.03}$ | $\mathbf{0.00 \pm 0.05/0.06}$ | $\mathbf{0.00 \pm 0.03/0.06}$ | $\mathbf{0.01 \pm 0.05/0.31}$ |

achieves perfect orthogonality. However, as we move to models with higher output dimensions, this near-orthogonality property diminishes significantly. The higher the output dimension, the less likely it is for the manifold associated with a good task loss to overlap with the manifold exhibiting near-orthogonality.

## 5 SUMMARY

### 5.1 RETHINKING STRICT ORTHOGONALITY REGULARIZATION

In response to the question posed in the introduction: While it is possible to maintain strict orthogonality, the kernel matrix $\boldsymbol{KK}^\top$ may not always be the ideal entity for this orthogonality. Our observations on near-orthogonality, as presented in Table 2, coupled with the enhancement in model performance depicted in Table 1, indicate that strict orthogonality regularization might not always be the most effective strategy.

Take, for instance, the ResNet56, as referenced in both Table 2 and Table 1. Although this model can achieve near-orthogonality during its training phase and also benefit from strict orthogonality regularization in terms of output dimension, it doesn't perform as well when compared to the relaxed orthogonality regularization applied on the transition dimension. This highlights a contradiction between the model's increased capacity and the strict orthogonality within output space.

For models characterized by heightened expressivity, typically associated with deeper architectures, it is crucial to exercise judiciousness when enforcing strict orthogonality on the kernel matrix.

### 5.2 LIMITATIONS

Our methodology presents certain limitations that warrant further exploration and potential solutions.

- *Estimation of Transition Dimension*: The current method for estimating the transition dimension could be enhanced with a more theoretically robust approach. In an ideal scenario, we would design models that facilitate module-wise relaxation configurations, providing greater adaptability and refined oversight during the training process.

- *Computational Complexity*: Our relaxed orthogonality regularization introduces a relatively large computational challenge. Specifically, the integration of a dual search mechanism within correlation matrix, targeting both positive and negative boundaries of the relaxation correlation filter pair, implies that the relaxed orthogonality regularization is typically more resource-demanding than its strict counterpart. Exploring ways to reduce the computational demands of relaxed orthogonality regularization warrant further research.

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

## A  APPENDIX

### A.1  IMPLICATIONS OF TASK LOSS ON ORTHOGONALITY REGULARIZATION

In this section, we discuss the interplay between task loss and strict orthogonality, particularly within what we term as the output space. This space, defined by the black coordinate filters, encompasses the transition dimension, which is of particular interest to our study.

Consider the output space, spanned by the black orthogonal filters. Within this space, the green-shadowed transition dimension is embedded, which is effectively spanned by a set of linear filters $\{k_1, k_2, \ldots, k_n\}$. However, due to the higher dimensionality of the out space compared to the transition dimension, we observe "redundant" transition dimension filters, such as $k_{n+1}$.

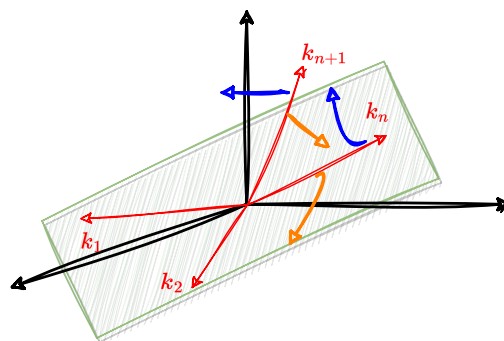

Figure 3: Depiction of the conflict between strict orthogonality and task loss

During optimization, a conflict arises between the structural filters of the transition dimension and the task filters. Let's delve deeper into this conflict by examining two specific filters, $k_n$ and $k_{n+1}$:

- From the perspective of strict orthogonality regularization, purple gradients are imposed that strive to span a larger transition dimension. This results in the extraction of in-span filter $k_n$ from the transition dimension and the orthogonalization of $k_{n+1}$, regardless of the existing linear span of the transition dimension.

- On the other hand, the task loss introduces orange gradients on $k_n$ and $k_{n+1}$, possibly drawing filters into the current transition dimension. Simultaneously, it rearranges the filter distribution within the transition dimension to optimize the layer-wise data representation.

When focusing solely on strict orthogonality in the output dimension, issues arise if the purple gradients become too strong, leading to over-regularization of orthogonality. This may inadvertently result in a wastage of filters, given that the existing linear span of the transition dimension can effectively represent most of the input.

Our proposed resolution involves relaxing some highly correlated pairs like $(k_n, k_{n+1})$ from the correlation orthogonality regularization. By doing so, we believe that such relaxation can assuage the conflicts in the imagined transition dimension and strike a balance between orthogonality and task performance.

### A.2  GRADIENT ANALYSIS OF STRICT ORTHOGONALITY REGULARIZATION

Given the holistic entity kernel orthogonality regularization in Frobenius norm, this section aims to provide a deeper understanding of the regularization term and its implications in training. We can break down the regularization term into two parts: one for the correlation term and one for the diagonal term. This decomposition allows us to separately analyze the contributions of off-diagonal and diagonal elements to the regularization.

$$\mathcal{L} = \left\| (\boldsymbol{K}\boldsymbol{K}^\top - \boldsymbol{I}) \right\|_F^2$$
$$= \left\| \text{off-diagonal}(\boldsymbol{K}\boldsymbol{K}^\top - \boldsymbol{I}) \right\|_F^2 + \left\| \text{diag}(\boldsymbol{K}\boldsymbol{K}^\top - \boldsymbol{I}) \right\|_2^2 \qquad (16)$$
$$= \left\| \text{off-diagonal}(\boldsymbol{K}\boldsymbol{K}^\top - \boldsymbol{I}) \right\|_F^2 + \lambda \left\| \text{diag}(\boldsymbol{K}\boldsymbol{K}^\top - \boldsymbol{I}) \right\|_2^2 (\lambda = 1)$$

We will discuss the reason why insert a scaling factor $\lambda$ in our strict orthogonality regularization later in this section. The off-diagonal matrix, for correlation part $\boldsymbol{C}$ which contains only the off-diagonal entries of $\boldsymbol{K}\boldsymbol{K}^\top - \boldsymbol{I}$:

$$\boldsymbol{C} = \boldsymbol{K}\boldsymbol{K}^\top - \boldsymbol{I} - \text{diag}(\boldsymbol{K}\boldsymbol{K}^\top - \boldsymbol{I}) \qquad (17)$$

The Frobenius norm of $\boldsymbol{C}$ and its gradient with respect to $\boldsymbol{K}$ are:

$$\|\boldsymbol{C}\|_F^2 = \text{trace}(\boldsymbol{C}^\top \boldsymbol{C}) \quad \text{and} \quad \nabla_{\boldsymbol{K}}\|\boldsymbol{C}\|_F^2 = 2(\boldsymbol{K}\boldsymbol{C} + \boldsymbol{C}\boldsymbol{K}^\top) \qquad (18)$$

For the diagonal term, let's define the diagonal matrix $\boldsymbol{D}$:

$$\boldsymbol{D} = \text{diag}(\boldsymbol{K}\boldsymbol{K}^\top - \boldsymbol{I}) \qquad (19)$$

The Euclidean norm of $\boldsymbol{D}$ and its gradient with respect to $\boldsymbol{K}$ are:

$$\|\boldsymbol{D}\|_2^2 = \text{trace}(\boldsymbol{D}^\top \boldsymbol{D}) \quad \text{and} \quad \nabla_{\boldsymbol{K}}\|\boldsymbol{D}\|_2^2 = 2\boldsymbol{K}\boldsymbol{D} \qquad (20)$$

Combining the gradients, the total gradient of $\mathcal{L}$ with respect to $\boldsymbol{K}$ is:

$$\nabla_{\boldsymbol{K}}\mathcal{L} = 2(\boldsymbol{K}\boldsymbol{C} + \boldsymbol{C}\boldsymbol{K}^\top) + 2\lambda\boldsymbol{K}\boldsymbol{D} \qquad (21)$$

This gradient can be used in optimization algorithms to update the matrix $\boldsymbol{K}$ and minimize the loss function $\mathcal{L}$. However, it's essential to note that from Appendix A.1 and Table 2, we know due to the task loss, the kernel matrix cannot be an identity matrix. Furthermore, the residual in the diagonal part $\boldsymbol{D}$ will be larger than the residual in the correlation part $\boldsymbol{C}$. As the initial motivation of the orthogonality regularization is to alleviate filters redundancy, we think it natural to focus the gradinet on the correlation part, by adjusting the $\lambda$ coefficient.

### A.3 On the inherent non-compliance of specific convolutional layers with the kernel orthogonality definition

Consider an overdetermined matrix $\mathbf{K}$ of dimension $o \times (i \times k_h \times k_w)$ where $o > i \times k_h \times k_w$. We aim to prove that the kernel function $\mathbf{K}\mathbf{K}^\top$ cannot be equal to the identity matrix $\mathbf{I}_{o \times o}$:

The rank of $\mathbf{K}\mathbf{K}^\top$ is at most the rank of $\mathbf{K}$. Since $\mathbf{K}$ has more rows than columns (due to the constraint $o > i \times k_h \times k_w$), its rank is bounded by its column count, i.e., $\text{rank}(\mathbf{K}) \leq i \times k_h \times k_w$. Consequently, $\text{rank}(\mathbf{K}\mathbf{K}^\top) \leq i \times k_h \times k_w$.

For $\mathbf{K}\mathbf{K}^\top$ to be the identity matrix $\mathbf{I}_{o \times o}$, the rows of $\mathbf{K}$ must be orthonormal. This condition necessitates that the dot product of any pair of distinct rows is zero.

However, given the dimensionality constraint $o > i \times k_h \times k_w$, it is evident that not all rows of $\mathbf{K}$ can be orthonormal. This is because the maximum number of linearly independent rows a matrix with $i \times k_h \times k_w$ columns can possess is $i \times k_h \times k_w$. As $o > i \times k_h \times k_w$, there are more rows in $\mathbf{K}$ than the maximum permissible number of linearly independent rows. This ensures the existence of at least one pair of rows that are linearly dependent, contradicting the orthonormality requirement.

In conclusion, given the specified dimensions of $\mathbf{K}$, $\mathbf{K}\mathbf{K}^\top$ cannot be $\mathbf{I}_{o \times o}$.

### A.4 Analysis of Norm Constraints in Overdetermined Convolutional Filters

We enforce an orthogonality structure on $n$ filters in the overdetermined convolutional filters $\boldsymbol{K}$. If we further enforce the diagonal to be unitary, an identity submatrix $\boldsymbol{I}_n$ will be present in $\boldsymbol{K}$. In the following section, we delve into the filter norm. Given that the index of the feature is inconsequential for our discussion, we can, for the sake of research simplicity, rearrange the rows of $\boldsymbol{K}$ to form the orthogonal structure with unitary norms $\boldsymbol{I}_n$. When $\boldsymbol{x}$ is multiplied by $\boldsymbol{I}_n$ in $\boldsymbol{K}$, the components of $\boldsymbol{x}$ remain unchanged.

Beyond the identity submatrix, the matrix $\boldsymbol{K}$ possesses additional rows. These rows, when multiplied with $\boldsymbol{x}$, will produce additional components in $\boldsymbol{K}\boldsymbol{x}$. The squared norm of $\boldsymbol{K}\boldsymbol{x}$ will increase due to the contribution from these additional components.

The squared norm of $\boldsymbol{K}\boldsymbol{x}$ is expressed as:

$$\|\boldsymbol{K}\boldsymbol{x}\|^2 = \boldsymbol{x}^\top \boldsymbol{K}^\top \boldsymbol{K} \boldsymbol{x}$$

Due to the presence of $\boldsymbol{I}_n$ in $\boldsymbol{K}$, the contribution to the squared norm becomes $\boldsymbol{x}^\top \boldsymbol{x}$, equivalent to the squared norm of $\boldsymbol{x}$. The squared norm contribution from the remaining components will always be non-negative, as squared norms are inherently non-negative.

Given the inclusion of the identity submatrix $\boldsymbol{I}_n$ in $\boldsymbol{K}$, it is evident that the norm of $\boldsymbol{K}\boldsymbol{x}$ will always be greater than or equal to the norm of $\boldsymbol{x}$. Formally:

$$\|\boldsymbol{K}\boldsymbol{x}\| \geq \|\boldsymbol{x}\|$$

The condition of equality will hold true if the additional rows in $\boldsymbol{K}$ yield zero vectors upon multiplication with $\boldsymbol{x}$. However, in practice, the existence of the identity submatrix ensures that the norm $\|\boldsymbol{K}\boldsymbol{x}\|$ surpasses $\|\boldsymbol{x}\|$.

In practice, the cooperation of full orthogonality regularization cause the overdetermined filters amplifying the magnitude of the input signal. This behavior contradicts the design principle of overdetermined filters, which are intended to function similarly to skip connections.

