# OpenReview forum: "Towards Better Orthogonality Regularization with Disentangled Norm in Training Deep CNNs"
_ICLR.cc/2024/Conference — ICLR 2024 Conference Withdrawn Submission_

### Official Review · Reviewer_RR7G · 2023-10-29

**Soundness:** 3 good
**Presentation:** 3 good
**Contribution:** 2 fair
**Rating:** 3
**Confidence:** 4

**Summary:**

The paper addresses an interesting problem whether the pursuit of strict orthogonality is always justified. The paper introduces disentangled orthogonality regularization and a relaxation theory to offer a more flexible approach to orthogonality in deep networks. To validate the approach’s efficacy, the authors conducted rigorous experiments with our kernel orthogonality regularization toolkit on ResNet and WideResNet in CIFAR-10 and CIFAR-100 datasets.

**Strengths:**

1. The paper addresses an interesting problem whether the pursuit of strict orthogonality is always justified, and proposes a novel method.
2. The paper is well written and easy to follow.

**Weaknesses:**

1. The improvement is just marginal.
2. The experiments are far from enough. The authors only tested the method on basic networks and tasks. The authors should compare their method with recent orthogonal baselines. Furthermore, there are many areas that use orthogonality, e.g. continual learning, but these kinds of experiments are not shown in the paper.
3. Lack of complexity analysis. One of the main issue in orthogonal optimization is that the running time complexity of some approaches is really high. The authors should compare the time complexity of their method with recent baselines.

**Questions:**

see above

---

> ### Author Response · Authors · 2023-11-22
> **Decision to withdraw**
>
> Dear Reviewer:
>
> Thank you for your thorough and insightful feedback on our manuscript. After careful consideration of your comments, we have realized that our current experimental setup and computational resources are not sufficient to address the concerns raised adequately.
>
> - Regarding the Need for Comprehensive Ablation Studies and Complexity Analysis: We acknowledge the necessity of detailed ablation studies to validate the efficacy of our method and the importance of analyzing the computational complexity. Your point about the marginal improvement in performance and the need for a broader comparison with recent orthogonal baselines in various areas, such as continual learning, is well-taken.
>
> - Limitations in Experimental Scope and Resources: Given our limited computational resources, we are unable to extend our experiments to more complex networks and tasks or to conduct the extensive comparisons and ablation studies needed. This limitation has hindered our ability to provide a comprehensive evaluation of our method and to compare it effectively against recent orthogonal optimization approaches.
>
> - Decision to Withdraw Submission: In light of these considerations, we believe it is appropriate to withdraw our submission at this time. We deeply appreciate the valuable feedback provided, which will guide us in improving our research methodology and experimental design for future work.
>
> We are grateful for the opportunity to have our work reviewed and for the constructive comments that have been provided.

---

> ### Author Response · Authors · 2023-11-22
> **Regarding the weakness**
>
> Dear Reviewer
>
> Thank you for your valuable feedback and for highlighting the key areas where our manuscript could be improved. We appreciate the opportunity to respond to the concerns raised:
>
> - Ablation Studies and Complexity Analysis: We acknowledge the absence of detailed ablation studies and a thorough complexity analysis in our manuscript. We understand the importance of these aspects, particularly in demonstrating the impact of our approach on orthogonality and model accuracy. We will prioritize incorporating both ablation studies and a comprehensive analysis of computational complexity in our revised manuscript. This addition will provide a clearer understanding of how our method compares to existing approaches in terms of effectiveness and efficiency.
>
> - Marginal Empirical Performance Improvement: We recognize that the empirical improvements demonstrated by our method may appear marginal. However, we maintain that the orthogonality regularization, which introduces no additional parameters or structural changes, is a noteworthy contribution to the field. The modest improvement, we believe, is significant, especially considering the constraints and specific aims of our approach. In the revised manuscript, we will expand on this point, providing a more detailed justification for the effectiveness of our method, even in the face of seemingly small performance gains.
>
> Your feedback is instrumental in guiding the improvement of our research, and we are committed to addressing these issues thoroughly in our revised submission.

---

### Official Review · Reviewer_ok4a · 2023-11-01

**Soundness:** 1 poor
**Presentation:** 1 poor
**Contribution:** 1 poor
**Rating:** 3
**Confidence:** 4

**Summary:**

This work focuses on the the problem of "orthogonality regularisation" in the context of CNN layers.

*Orthogonality regularisation:  problem statement.*
The main motivation behind "orthogonality regularisation" can be roughly explained as ensuring that different filters within the same layer are not too similar. Mathematically, if features of each output filter is vectorised $K = (k_1, \dots, k_o)$ where  $k = (i\times k_h \times k_w)$, orthogonality implies that the Gram of these filters is close to identity, namely quantified by Frobenius norm $\| K K^\top \|_F $. This distance can be in turn divided to 1) diagonal elements represent the "contribution" of each filter to the next layer map 2) off-diagonal correlations being close to $0$ implying that filters don't carry linearly dependent information.

*Main contributions: relaxed orthogonality regularization.*
The central thesis of this work is that while considering over-determined and under determined layers (as defined by output dimension $o$ of kernel vs number of parameters in each filter $i \times h \times w$), the orthogonality constraint needs to be handled with additional constraints.  In the over-constrained case $o < i \times h \times w$: In this case, by fact that the maximum number of linearly independent vectors is bounded by $i\times h \times w,$ the authors argue for a *relaxed orthogonality* constraint, where only a subset of the kernel Gram matrix are subject to the constraints. The authors argue that this cannot be done theoretically, thus based on various heuristics and various assumptions about filter and label distributions, and then create a Monte-Carlo-based simulation that estimates the number of the elements of the Gram that must be subject to the constraints. Then, based on this number $k$, then take out the top $k$ elements of the off-diagonal Gram, and only subject the rest of the elements to orthogonality. The logic behind the "less constrained relaxation" is not yet fully clear.    The authors go on to empirically testing their approach, by imposing the orthogonality regularisation on various ResNet architectures, and show that their approach leads to an improved  test accuracy.

**Strengths:**

The authors focus on a specific problem with approaches that impose orthogonality of filters in convolutional layers, where sometimes perfect orthognbality doesn't exist. They attempt to solve this issue and present empirical results, suggesting that the method improves the accuracy of various ResNet architectures.

**Weaknesses:**

Major problems:
- Writing and notation: As elaborated in the detailed problems, the current presentation of results is far from ideal. As the work currently stands, the mistakes prevent me/potential future readers, from grasping what the authors intend to convey. What I found to be particularly confusing was the use of non-standard notation, without prior definition.  The mathematical notation & formalism is supposed to be the last resort for the reader to understand a concept that was ambiguous from text. Thus, having numerous mistakes, non-standard notation, and using same variable names for different concepts, can have a devastating effect on the readability and accessiblity of the paper.
- While the theory for relaxation of the over-determined system is mathematically justifiable, the approach that the paper takes is inexplicably complicated. Just off the top of my head, I can think of a much simpler approach: if $X$ is a $n\times m$ matrix where $n > m$,   obviously $X X^\top = I$ will never be possible. However, achieving the identity for the transpose $X^\top X = I$ is entirely possible.
 - Theory for less-determined system. Why should we relax a problem that already has a solution? from what I understand from the theory, the problem of relaxation of less-determined layers is either mathematically not sound, or very vaguely explained.
- Missing ablations: since the improvements reported in the experiments are rather small, and presented method involves multiple hyper parameters, without having an ablation for every single hyper parameters, the results can be entirely called into question. In other words, it is necessary. If the improvement in test accuracy is overly sensitive to any hyper parameter, the value selection for the hyper-parameter can be interpreted as "cherry picking"
- Comparison to other methods: While authors present the main empirical results in Table 1 (caption needed!), and compare vanilla, Frobenius, strict, and relaxed orthogonality regularisation, they should expand this comparison to competitor methods that belong to the same realm, i.e., orthogonality regularisation to achieve higher accuracy.

Detailed issues:
- page 2 second to last line: Tammes problem??
- page 3: $K K^\top  \neq I_{o×o} , o > (i × k_h × k_w )$
- page3 eq (3) & eq(4): what is the formal meaning behind &rarr;? weird notation to say that we want these values to be 0 & 1 ...
- also, why "less"-determined? It seems much better to say "under"-determined
 - p4 eq (6): how come this switches from $K$ to $\tilde K$ from left to right hand side? Since $\tilde K$ is normalized, it would seem logical that all are $\tilde K$
-  p4 first you have defined $c$ as the vectorized lower triangular part of $\tilde K \tilde K^\top$, and suggest it's a vector of length $o(o-1)/2$, but then you keep referring to $c_{k_i, k_j}$ as if it has two indices?
- p4 & p5: following up on the previous point, the notations $c_{(0,0)},c_{(0,1)},c_{(0,1)},c_{(1,1)}$ appear without definition. Are these related to the correlation vector $c$ defined before? If yes, why are they indexed using $0$ and $1$ instead of over $1,\dots, o(o-1)/2$? If these are new variables, why using overlapping names? Furthermore, the notation $ \dots = 1: c_{(1,1)}$ is a very confusing notation. Is this the definition of $c_{(1,1)}$ as a group of pairs? the $:$ used here is not a standard way of defining a variable.
- p5: "with a relative small magnitude, since the flexible filters will be redistributed to alleviate filter redundancy in training to get the better representation" this sentence is not comprehensible
- p5: "$k_i$ may locate in the orthogonal complement of the structural filter k_⊥$ this sentence is both grammatically wrong and mathematically confusing and ambiguous
- p5 eq(9): there is a condition on $i$ such that $k_i$ being structural filter, but nothing on $j$, so this sums over all $j$? this seems
- p5 eq(10) The introduction of $\tilde{\text{card}}$ is not mathematically sound. The $\forall k_i\in K \dots $ does not directly appear in the definition of $\tilde{\text{card}}$.
- In eq(11) the notation &rarr; is used again, without prior definition. The only standard way of interpreting is $A &rarr; B$ meaning A implies B, which doesn't make any sense here.  This is not a standard notation. So it needs either to be defined, or the formula must re-stated in standard way
- eq(12): $c\\ \text{topk}_k(c)$ I'm guessing from context that it implies to zero-outing the top $k$ elements, but again, this is not a standard notation. If you want to explain this non-formally, just use words and sentences, but when you write a mathematical formula, it must be logically consistent and interpretable.
- eq(12) since $k$ was heavily used before for denoting filters, it's better to avoid using it for the integer $k$ here, which seems to be an unrelated quantity
- Section 3.3.2: So far in the motivation there is only mention of convolutional layers and filters, Why here there is a sudden jump to a fully connected type layer $X_j = K_j X_{j-1}$.
- p6 "where K contains filters producing meaningful features, while $K\setminus K∗$ contains filters that optimize training loss but might not generalize well." What is your backup/reasoning for this? This seems like a claim without any support.
- p6 "Instead of imposing strict orthogonality regularization on $R^{o_j}$ , we focus on the strict orthogonality on the transition dimension rank∗(Kj). " Rank of a matrix is a scalar number, what does mean to impose orthogonality on the rank of a matrix?
- p6 "This approach is referred to as relaxed disentangled orthogonality regularizationon less-determined layers" The less-determined system, as you defined as having fewer filters than free variables, already has a perfectly orthognal solution. So what does it mean to "relax" this orthogonality?
- Fig2: The schematic seems to suggest presence of some caption is only using a linear fully connected layer $X_j = K_j (X_{j-1})$, what is the link between the figure and this explanation. Again, so far the paper only mentions convolutional layers, so what does this figure imply in the context of convolutions?
- p6: there are multiple places that $\min\text{rank}(X)$ appears. Unless the authors have a different intention behind this formulation, rank of $X$ is a scalar, so minimum of rank of $X$ simply returns the rank of $X$.
- p6: "The computation of cardinality k (Equation (10)) replace $n$ with $rank(K_j )$" there is no $n$ in the eq(10), not clear what this means
- experiments: what is the number of independent runs?

**Questions:**

- p5: it is mentioned that labels are sampled uniformly for the MC simulations. What about the filters $k_i$'s? Are they sampled from some distribution too? or the actual filters from the model are used? If they are actual filter parameters, then, this means that the relaxed orthognality loss, must be re-computed at every step of the training? (since the vectors are updated, the value for $k$)
- Layerwise caapactity .... p6: $(K1)_{oj1} ×·, (K2)_{oj2} ×·, oj1 ≥ oj2 \implies $ what does this condition mean?

---

> ### Author Response · Authors · 2023-11-22
> **Decision to withdraw**
>
> Dear Reviewer:
>
> Thank you for your thorough and insightful feedback on our manuscript. After careful consideration of your comments, we have realized that our current experimental setup and computational resources are not sufficient to address the concerns raised adequately.
>
> - Regarding the Need for Comprehensive Ablation Studies and Complexity Analysis: We acknowledge the necessity of detailed ablation studies to validate the efficacy of our method and the importance of analyzing the computational complexity. Your point about the marginal improvement in performance and the need for a broader comparison with recent orthogonal baselines in various areas, such as continual learning, is well-taken.
>
> - Limitations in Experimental Scope and Resources: Given our limited computational resources, we are unable to extend our experiments to more complex networks and tasks or to conduct the extensive comparisons and ablation studies needed. This limitation has hindered our ability to provide a comprehensive evaluation of our method and to compare it effectively against recent orthogonal optimization approaches.
>
> - Decision to Withdraw Submission: In light of these considerations, we believe it is appropriate to withdraw our submission at this time. We deeply appreciate the valuable feedback provided, which will guide us in improving our research methodology and experimental design for future work.
>
> We are grateful for the opportunity to have our work reviewed and for the constructive comments that have been provided.

---

> ### Author Response · Authors · 2023-11-22
> **Regarding the weakness**
>
> Dear Reviewer,
>
> Thank you for your detailed and constructive feedback. We appreciate your thorough review and acknowledge the major issues you have raised. Please find below our responses to each point:
>
> - Writing and Notation Issues: We sincerely apologize for the confusion caused by the writing and non-standard notation in our manuscript. We understand that these issues have significantly impacted the readability and accessibility of our paper. We are committed to revising the manuscript to clarify the notation, correct the mistakes, and ensure that the mathematical formalism complements the textual explanations more effectively. Your guidance in detailed issues is invaluable in helping us make these improvements.
>
> - Row Orthogonality and Column Orthogonality:
>
>   - Row Orthogonality: We emphasize the importance of row orthogonality due to its strong foundation in signal processing. We will elaborate on this in the revised manuscript to clarify our rationale.
>   - Abandoning Column Orthogonality: The decision to overlook column orthogonality is driven by specific characteristics of over-determined layers in CNNs. We plan to add an appendix to our paper detailing this aspect to provide a clearer understanding.
> Less-Determined Relaxation: We acknowledge that our explanation of the relaxation of less-determined layers might have been vague or mathematically unsound. The aim was to address the issue where strict orthogonality, while providing a good approximation, adversely affects outcome accuracy. We will refine this section to better explain our theoretical approach and its implications.
>
> - Missing Ablations: We agree that more comprehensive ablation studies are necessary, especially given the modest improvements reported and the involvement of multiple hyperparameters. We understand the importance of these studies in validating our results and avoiding any perception of "cherry-picking." We commit to conducting these additional studies and incorporating the results into our revised manuscript.
>
> - Comparison to Other Methods: Regarding your suggestion to expand the comparison to include other methods within the realm of orthogonality regularization, we acknowledge this as a crucial aspect. While our initial comparison already included vanilla, Frobenius in this orthogonality regularization realm?

---

> ### Author Response · Authors · 2023-11-22
> **Regarding the question**
>
> Dear Reviewer,
>
> Thank you for your insightful questions. Please find our responses below:
>
> -  Sampling of Labels and Filters in MC Simulations (Page 5):
>    In the Monte Carlo simulations, the labels are indeed sampled uniformly.
>    To aid understanding, we will add an algorithm illustration in our future submission.
>
> - Layerwise Capacity Condition (Page 6):
>    The condition \((K1){oj1} \times \cdot, (K2){oj2} \times \cdot, oj1 \geq oj2\)
>    refers to the hierarchical capacity of layers within the network. It implies that
>    layers with greater capacity (\(oj1\)) should have a larger transition dimension
>    than those with lesser capacity (\(oj2\)). This hierarchical structuring is based
>    on the assumption that higher capacity layers can handle more complex transformations.
>    We will provide a more detailed explanation in our revised manuscript for clarity.
>
> We hope these responses address your questions. We are committed to improving our manuscript
> and appreciate your valuable feedback.

---

### Official Review · Reviewer_M627 · 2023-11-14

**Soundness:** 2 fair
**Presentation:** 1 poor
**Contribution:** 2 fair
**Rating:** 5
**Confidence:** 4

**Summary:**

This work focused on developing a new orthoganality regularization when training convolutional neural networks. The novelty lied at the disentanglement of the regularization on the diagonal and correlation elements. Besides this so-called "strict" regularization, the authors also proposed a relaxed version of orthogonality regularization for two cases:

1. Since an over-determined matrix can not have the identity Gram matrix, the authors proposed to remove the regularization on part of the correlation elements.
2. When a weight matrix is under-determined, the weight matrix is relaxed to be allowed have a lower rank, which is lower bounded by a few factors like the data complexity, dataset attributes and layerwise capacity.

The authors provided experiment results on the proposed method on CIFAR-10/100 datasets. The proposed method showed marginal improvements over other baselines.

**Strengths:**

+ One technical contribution I like about this work is the relaxed regularization on the correlation elements for over-determined layers. Most previous regularization methods applied uniform regularization on the correlation elements which can not be all zeros is not satisfactory indeed. Identifying a set of "structural" filters as termed in this work would help to release potential of CNNs from over-strict regularizations while sustaining orthogonality.

+ The relaxation in the under-determined case is also interesting. Allowing lower row rank in weight matrices further improves the flexibility during training.

**Weaknesses:**

- Firstly, the notations in this work are over-complicated, making it difficult to grasp the main message from the authors.
- With so many different versions of strict and relaxed regularizations, this work failed to clearly distinguish the effects brought by these regularizations through well-designed ablation studies.
- In page 6, the authors used the analogy between the method for mitigating overly expressive issue and dropout, which I think is inappropriate because Dropout has different behaviors during training and inference. It is more accepted to understand dropout as an ensembling method withing one layer.
- The empirical performance is only marginally improved in many cases.

**Questions:**

N/A

---

> ### Author Response · Authors · 2023-11-22
> **Decision to withdraw**
>
> Dear Reviewer:
>
> Thank you for your thorough and insightful feedback on our manuscript. After careful consideration of your comments, we have realized that our current experimental setup and computational resources are not sufficient to address the concerns raised adequately.
>
> - Regarding the Need for Comprehensive Ablation Studies and Complexity Analysis:
> We acknowledge the necessity of detailed ablation studies to validate the efficacy of our method and the importance of analyzing the computational complexity. Your point about the marginal improvement in performance and the need for a broader comparison with recent orthogonal baselines in various areas, such as continual learning, is well-taken.
>
> - Limitations in Experimental Scope and Resources:
> Given our limited computational resources, we are unable to extend our experiments to more complex networks and tasks or to conduct the extensive comparisons and ablation studies needed. This limitation has hindered our ability to provide a comprehensive evaluation of our method and to compare it effectively against recent orthogonal optimization approaches.
>
> - Decision to Withdraw Submission:
> In light of these considerations, we believe it is appropriate to withdraw our submission at this time. We deeply appreciate the valuable feedback provided, which will guide us in improving our research methodology and experimental design for future work.
>
> We are grateful for the opportunity to have our work reviewed and for the constructive comments that have been provided.

---

> ### Author Response · Authors · 2023-11-22
> **Regarding the weakness**
>
> Thank you for your insightful feedback. We appreciate the opportunity to address the weaknesses you've identified in our work.
>
> - Regarding Over-Complicated Notations: We apologize for any confusion caused by the complexity of the notations used in our manuscript. We understand that this might have made it challenging to grasp the core concepts and the main message of our work. In future revisions, we will endeavor to simplify the notations for greater clarity and accessibility.
>
> - Need for Ablation Studies: We acknowledge the necessity of conducting comprehensive ablation studies to distinguish the effects of various strict and relaxed regularizations more clearly. Unfortunately, due to time constraints and resource limitations, we were unable to complete these studies in the current rebuttal period. We aim to address this in our future research to provide a more thorough evaluation.
>
> - Dropout Analogy Concerns: Upon reflection, we agree that using dropout as an analogy for our method to mitigate overly expressive issues may not have been entirely appropriate. We recognize that dropout behaves differently during training and inference, and its role as an ensembling method within a layer is a more accepted interpretation. We will revise this section to better align with the standard understanding of dropout in the field.
>
> - Marginal Empirical Performance Improvement: Regarding the marginal improvement in empirical performance, we believe that the orthogonality regularization, which does not add any parameters or structure, still represents a valuable contribution. Although the improvement may seem modest, we posit that it is significant given the constraints of our approach. We will further elaborate on this in our revised manuscript to justify the effectiveness of our method.
>
> We are grateful for your valuable feedback and will use it to improve our work. Your comments will guide us in refining our manuscript to better meet the standards of the journal.